# Health care professionals' attitudes, behaviours and barriers toward exercise promotion among patients: A systematic review

Pinar Avsar[1,2,3]*, Zena Moore[1,2,4,5,6,7,8], Husain Nasaif[9], Barry Moore[10], Declan Patton[1,2,4,5,7], Tom O'Connor[1,2,4,5,7,11], Vishnu Renjith[1,3,12]

1 Skin Wounds and Trauma Research Centre, RCSI University of Medicine and Health Sciences, Dublin, Ireland, 2 School of Nursing and Midwifery, RCSI University of Medicine and Health Sciences, Dublin, Ireland, 3 Cardiff University, Cardiff, United Kingdom, 4 Fakeeh College of Health Sciences, Jeddah, Saudi Arabia, 5 School of Nursing and Midwifery, Griffith University, Queensland, Australia, 6 Faculty of Medicine, Nursing and Health Sciencses, Monash University, Melbourne, Australia, 7 Department of Public Health, Faculty of Medicine and Health Sciences, Ghent University, Ghent, Belgium, 8 School of Nursing and Midwifery, Curtin University, Perth, Australia, 9 RCSI Bahrain, Adliya, Kingdom of Bahrain, 10 Curtin University, Perth, Australia, 11 Faculty of Science, Medicine and Health, University of Wollongong, Wollongong, Australia, 12 Manipal College of Nursing, Manipal Academy of Higher Education, Manipal, India

* pinaravsar@rcsi.com

## Abstract

### Objective

To determine health care professionals' (HCPs) attitudes, behaviours, and barriers toward exercise promotion among patients.

### Method

Using systematic review methodology, we included published studies focusing on health care professionals' attitudes, behaviours, and barriers towards exercise promotion among patients. The search was conducted in June 2023, using the Cochrane Central Register of Controlled Trials (CENTRAL), PubMed/MEDLINE, Embase, Cumulative Index to Nursing and Allied Health Literature (CINAHL) Plus, and Scopus databases, and returned 352 records, of which 34 met the inclusion criteria. The Critical Appraisal Skills Programme (CASP) tool was used for quality appraisal.

### Results

Results revealed that HCPs hold a positive attitude towards advocating physical activity. Further physical activity was prescribed to achieve a number of objectives, including enhancing physical function, encouraging an active lifestyle, preventing complications, monitoring progress or compliance, and preventing functional decline. Barriers to exercise promotion identified related to four overarching categories: health

**Data availability statement:** All data are in the manuscript and/or Supporting information files.

**Funding:** The author(s) received no specific funding for this work.

**Competing interests:** The authors have declared that no competing interests exist.

**Abbreviations:** CASP, Critical Appraisal Skills Programme; HCP, Health Care Professionals; PA, Physical Activity; WHO, World Health Organisation; CENTRAL, Cochrane Central Register of Controlled Trials.

professional-based barriers, perceived patient-related barriers, organisational barriers and health system-related barriers.

## Conclusion

This review highlights positive attitudes of healthcare professionals toward promoting physical activity, as well as persistent barriers that hinder effective implementation. Addressing these barriers requires a comprehensive approach, including tailored support for healthcare professionals and improved organisational and systemic structures. Further research is needed to evaluate the effectiveness of strategies aimed at overcoming these obstacles and enhancing physical activity promotion.

## 1. Introduction

The benefits of physical activity (PA) and the consequences of inactivity on patients from different age groups are well-examined and established [1,2]. Indeed, evidence demonstrates that PA contributes to improving the health outcomes of patients with both acute and chronic health problems, including but not limited to, cancer, cardiovascular, respiratory, metabolic, musculoskeletal, and autoimmune diseases [3]. Additionally, physical activity (PA) has been demonstrated to improve patients' health-related quality of life, functional capacity, physical fitness, skin perfusion, and glycaemic control [4–6]. Furthermore, it contributes to weight reduction, alleviation of depression and fatigue, and mitigation of disease symptoms [7–9].

Healthcare Professionals (HCPs) have a key role in optimising the health status of patients and in promoting PA for healthy individuals and patients at risk of poor health outcomes due to disease morbidity and physical inactivity [4]. A recent umbrella review reported that the involvement of professionals from different disciplines is one of the factors that increases patients' adherence to PA [5]. However, some studies report that knowledge gaps and attitudes of HCPs could potentially influence the promotion of PA to patients [6]. Despite the significant role of HCPs in promoting PA, patients reported not having discussions about exercise promotion with HCPs. They indicated that the exercise promotion provided did not meet their needs, as it lacked effective exercise guidance [10].

HCPs, on the other hand, report being unfamiliar with guidelines relating to PA and that they had received no pre-registration education or formal training about exercise [7]. As such, evidence indicates that many HCPs have no prior training on how to plan and initiate conversations with older patients about PA [8]. Barriers to PA promotion by HCPs include lack of time, education, training, support structures, expert contact, and exercise programs [11,12]. Additional barriers are workloads, patient risk concerns, and lack of self-efficacy, awareness, and perceived competence [13].

Several studies have examined the attitudes, behaviours, and practices of HPCs toward promoting PA [9,14]. However, results show that the practices were inconsistent and varied across healthcare settings and different disciplines. To address this gap, it is essential to enhance understanding of HCPs' attitudes and behaviors

toward promoting PA within their areas of practice. This understanding could inform the integration of PA training into the curricula for HCPs and into their continuous professional development programs. To our knowledge, no systematic review has summarised what is known about the perception of HCPs' regarding PA promotion. A preliminary search in the PROSPERO systematic review registry and MEDLINE revealed no current systematic reviews or protocols on this topic. Therefore, this review aimed to summarize evidence regarding attitudes, behaviors, and barriers toward exercise promotion among healthcare professionals.

## 2. Aim and research question

The overall aim of this systematic review was to determine health care professionals' attitudes, behaviours, and barriers toward exercise promotion among patients.

### 2.1. Research question

The research question was formulated using the PICO (Population, Intervention, Comparison, Outcome) framework to ensure a structured and focused approach. The population of interest included all healthcare professionals. The intervention under consideration was any form of exercise promotion programme or plan, which was compared to the absence of such interventions. The primary outcomes were healthcare professionals' attitudes towards, behaviours related to, and barriers experienced in promoting exercise. Secondary outcomes included: (1) the instruments or tools used to measure attitudes and behaviours, and (2) the type or nature of exercise. Thus, the research question was: What are health care professionals' attitudes, behaviours and barriers toward exercise promotion among patients?

## 3. Methods

This systematic review was conducted in accordance with the Preferred Reporting Items for Systematic Reviews and Meta-Analyses (PRISMA) 2020 guidelines, and the completed PRISMA checklist is provided in the Supporting Information (See S1 Table).

### 3.1. Criteria for considering studies for this review

**Inclusion criteria.** Studies written in the English language, employing any study design, exploring health care professionals' attitudes, behaviours, and barriers towards exercise promotion among patients were eligible for inclusion.

**Exclusion criteria.** We excluded studies that did not involve an investigation of health care professionals' attitudes, behaviours and barriers towards exercise promotion among patients. In addition, studies not written in English, literature reviews and opinion papers were excluded.

### 3.2. Electronic searches

A comprehensive search was conducted across the following electronic databases from their inception to June 2023 using the Cochrane Central Register of Controlled Trials (CENTRAL), PubMed/MEDLINE, Embase, Cumulative Index to Nursing and Allied Health Literature (CINAHL) Plus, and Scopus

To identify additional relevant studies, including unpublished and ongoing work, the review also employed supplementary search strategies. These included screening the reference lists of all included studies and relevant reviews, searching grey literature sources via OpenGrey (www.opengrey.eu), and examining conference proceedings, research reports, and dissertations.

The keywords used in the search included:

#1 "healthcare professionals or healthcare workers or healthcare providers or physician or nurse or doctor"

#2 "barriers or attitudes or perceptions or opinions or thoughts or feelings or beliefs or behavior"

#3 "exercise or physical activity"

#4 #1 AND #2 AND #3

For the purpose of this study, HCPs were defined as individuals who provide medical care, support, and services to patients, including but not limited to physicians, nurses, doctors, and general healthcare workers. While terms like "physiotherapist" or "clinician" were not used as specific keywords, the broader category of "healthcare professionals" was selected to encompass a wide range of providers in various healthcare settings.

### 3.3. Study selection

The article titles were assessed by two authors independently (PA, HN), and abstracts were screened against the eligibility criteria. The full text of potentially relevant studies was then reviewed independently by two authors (PA, HN). A third reviewer (ZM) was involved to reach a consensus on the final corpus of included studies when discrepancies were identified between the two primary reviewers. PRISMA was adapted as a framework for reporting this SR. A PRISMA flow chart provides a visual display of literature flow, and the final corpus of citations included [15]. The study protocol was pre-registered with the International Prospective Register of Systematic Reviews (PROSPERO; CRD42022344777).

### 3.4. Data extraction

Data were extracted from included studies and inserted into a pre-designed data extraction table using the following headings: author, study year and country, setting, sample characteristics, study design, key findings, and quality appraisal result.

### 3.5. Data analysis

The findings were narratively summarised, providing an overview of the study setting, geographical location, setting, and sample characteristics. There was a mix of quantitative and qualitative studies in the included studies, and due to the heterogeneity in the reporting of analysis and results in the individual studies, a qualitative coding process was deemed most appropriate to collate and analyse the data as outlined by Renjith, Yesodharan [16]. This process involved breaking down the findings into small, meaningful units called codes. These codes were grouped based on shared concepts to form categories, which were then clustered into broader themes [17]. The resulting themes and categories are presented under the sections of attitudes, practices, and behaviors.

To minimise the risk of bias arising from the themes presented in the quantitative questionnaires, a multi-faceted approach was employed, as follows:

1. Data Sources: The data collection in the included studies incorporated a range of methodologies, including interviews, focus groups, and quantitative questionnaires. This triangulation allowed us to capture diverse perspectives, reducing the reliance on any single data source and thereby, enhancing the robustness of the findings.

2. Thematic Development and Validation: Themes emerging from the data were systematically identified through thematic analysis. These themes underwent rigorous review by the research team to ensure internal validity. Multiple reviewers independently examined the data to identify discrepancies and to resolve them through discussion, thus reducing individual bias in theme identification.

3. Reflexivity Practices: Reflexivity was actively employed throughout the analysis process. The research team engaged in ongoing critical reflection on how their own biases and preconceptions could influence the data interpretation. These reflections were documented and discussed to ensure transparency and rigor in the analysis.

### 3.6. Quality appraisal

The quality and risk of bias in the included studies were assessed using the Critical Appraisal Skills Programme (CASP) tool. CASP is a widely recognised tool used to critically appraise the design, conduct, and reporting of research studies.

Two independent reviewers evaluated the quality of all included studies, utilising the CASP checklists specific to cohort and qualitative studies. Any disagreements were resolved through discussion and consensus. For each study, the reviewers assigned a judgment of 'Yes,' 'No,' or 'Unclear' for each criterion. The judgments were based on the study information and the reviewers' expertise. Studies were categorised as high quality if they scored above 90% in the critical appraisal, moderate quality if above 60%, and low quality if 60% or below [10]. A summary of the CASP assessments for the included studies is presented in Table 3.

## 4. Results

### 4.1. Overview of all included studies

As shown in Fig 1, following reviews of titles & abstracts from a 352 total of citations, 310 were excluded. Next, a full-text review of the remaining citations resulted in a further four exclusions for the following reasons: not focusing on the primary outcome, and not a research paper (See Table 1). Finally, 34 studies were deemed to meet the inclusion criteria, and these form the basis of this systematic review [7–9,11–14,18–43].

**Description of the studies.** The supplementary document provides an outline of the data extracted from the included studies (See S2 Table).

**Geographical location.** A wide spread of countries is represented in the studies, with the USA having eight studies [12,17,21,23,25,30,36,38], and the UK having five studies [20,37,39,43,45]. Australia, Canada, Germany, Netherlands, and Saudi Arabia have two studies each [18–20,28,29,33,35,39–41]. The remaining studies were from Australia and New Zealand [31], Belgium [42], Brazil [7], Ireland [14], Italy [11], Korea [37], Singapore [13], Taiwan [43] and the UK and Ireland [8] and Wales [22]. One study included people from seven international oncology societies [27].

**Clinical settings.** The studies were conducted across a diverse variety of clinical settings, with the hospital setting representing 50% (17/34) of the studies and the community and/or primary care setting representing the next largest group (24%; 8/34). A further 18% (6/34) of studies included representatives from national and/or international societies, such as the Australian Cancer Clinical Membership Organisation [40], all physical therapists registered in New South Wales [39], the Cancer Nurses Society of Australia and the Cancer Nurses Section of the New Zealand Nurses Organisation [31], the Korean cancer association [37], and the Oncology Nursing Society [42]. As mentioned, one study [27] also included members of seven international oncology societies. The remaining 3 studies were conducted in long-term care, a psychological care setting and a spinal cord injury centre [9,18,20].

**Population and sample.** The largest group of participants, included in 47% (16/34) of the studies, were nurses, followed by multidisciplinary team members at 38% (13/34). Physiotherapists were included in two studies, and medical clinicians alone were included in tthree studies. The total number of participants included in the studies was 6,712 (Mean: 197; SD: ±247; min: 9; max: 956).

**Study designs and data collection methods.** A total of 59% (20/34) of studies employed a survey design, all of which used questionnaires to collect the data. A qualitative design was employed by 13 studies, with seven using individual interviews to collect the data, and five using focus group interviews to collect the data. One study used a Delphi method approach [21], while another employed a mixed methods approach using a questionnaire and individual interviews [37].

**Results for the outcomes.**

### 1. Attitudes to exercise promotion

Overall, 16 studies assessed the attitude of HCPs regarding physical activity. All studies used different scales or conceptual frameworks for measuring attitudes. Due to heterogeneity in the scales and measures, the findings are summarised descriptively. Seven studies focused on physical activity among cancer patients [11,14,27,35,37,40,41], three studies focused on the general population [18,22,25], two on geriatrics [8,21] and one each on primary care [24], community care [26], acute medical [36] and a liver transplant [38] population.

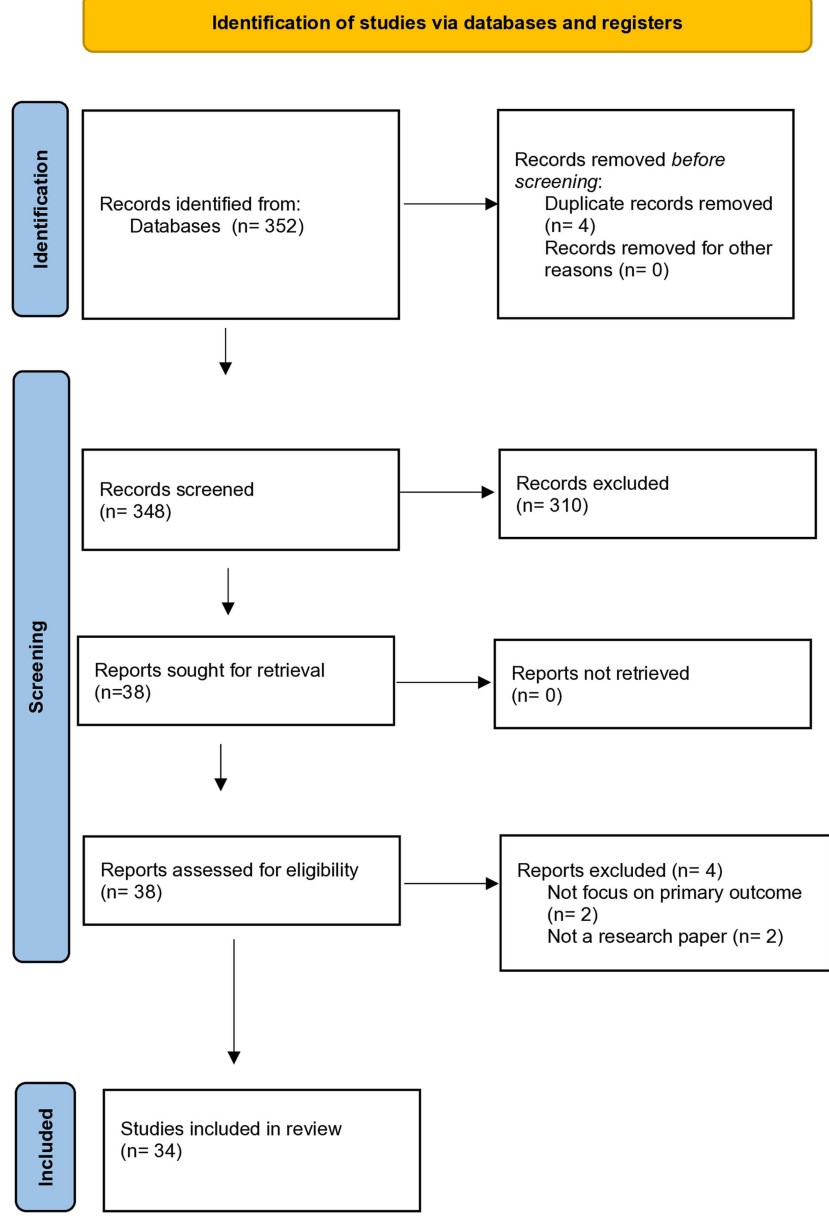

**Fig 1. PRISMA 2020 flow diagram.**

Overall, HCPs had a positive attitude towards the promotion of physical activity. In the studies focusing on physical activity among patients with cancer, HCPs identified both risks and benefits of physical activity. Furthermore, respondents highlighted the importance of undertaking a pragmatic approach based on the clinical condition and the individual ability of the patients [11,14,27,35,37,40,41]. Respondents also acknowledged that too much physical activity can negatively impact the health status of the patient [11]. However, in other studies, HCPs perceived that physical activity could promote health [25], improve quality of life [14,27], reduce fatigue [14,27], manage side effects of treatments [14] enhance behavioural changes [14] reduce cardiovascular risk [27]and improve activity tolerance [38].

**Table 1. Excluded studies.**

| Excluded studies with reasons for exclusion | | |
|---|---|---|
| **Author** | **Study** | **Reason for exclusion** |
| O'Donoghue, Cusack [44] | Contemporary undergraduate physiotherapy education in terms of physical activity and exercise prescription: practice tutors' knowledge, attitudes and beliefs | Not focusing on the primary outcome |
| Kontos, Alibhai [45] | A prospective 2-site parallel intervention trial of a research-based film to increase exercise amongst older hemodialysis patients | Not focusing on the primary outcome |
| Vishnubala and Pringle [46] | Working with healthcare professionals to promote physical activity | Not a research paper |
| van Hell-Cromwijk, Metzelthin [47] | Nurses' perceptions of their role with respect to promoting physical activity in adult patients: a systematic review | Not a research paper |

Recommending physical activity or exercise counselling was considered a priority and a part of the professional role [8,21,22,24,25,27,35–38,40,41]. Some respondents reported confidence in recommending physical activity to patients [27,40]; however, also reported that physical activity for patients should be recommended with caution [27,35].

## 2. Behaviours towards exercise promotion

Twenty-three studies reported on behaviours towards exercise promotion. Findings indicate that nurses and doctors were routinely involved in assessing the functional status of patients either during admission, while the patient was undergoing treatment or during discharge [28,36]. Exercise was recommended for four purposes: to promote physical function and an active lifestyle, to prevent complications, to monitor progress or compliance, and to prevent functional decline [12,17,39]. Additionally, the strategies employed to encourage physical activity among patients included assessment, advice, and counseling [39,40,44–46,48,49]; distribution of written information or pamphlets [40,49]; and referrals to specialists [39,40].

Nurses in four studies recognized their duty to promote ambulation and enhance patient physical activity based on health and circumstances [11,36,37,42]. Two studies [19,36] also found that nurses utilised referral services and signposted patients to physical activity services.

There were conflicting findings in relation to the prescription of physical activity among patients with cancer. In one study, the intention to promote physical activity in cancer patients was not strong [29] Conversely, in a second study physical activity for patients with cancer was recommended [21].

## 3. Barriers to exercise promotion

A total of 23 studies measured barriers to exercise promotion. The barriers identified were organised into four themes: 1: Health professional-based barriers, 2: Perceived patient-related barriers, 3: Organisational barriers and 4: Health system-related barriers. The overview of themes is represented in Fig 2.

1. Health professional-based barriers

Health professional related barriers included a lack of time to promote physical activity in routine work [11,12,14,20,30,34,35], and lack of awareness among HCPs regarding the type and benefits of physical activities [29,30]. Furthermore, five studies identified a lack of education and training programs on how to safely mobilise hospitalised patients as a barrier [11,18,21,27,32]. Patient safety concerns were also expressed as a barrier to the promotion of physical activity, such as the risk of additional injury and falls [7,12,35].

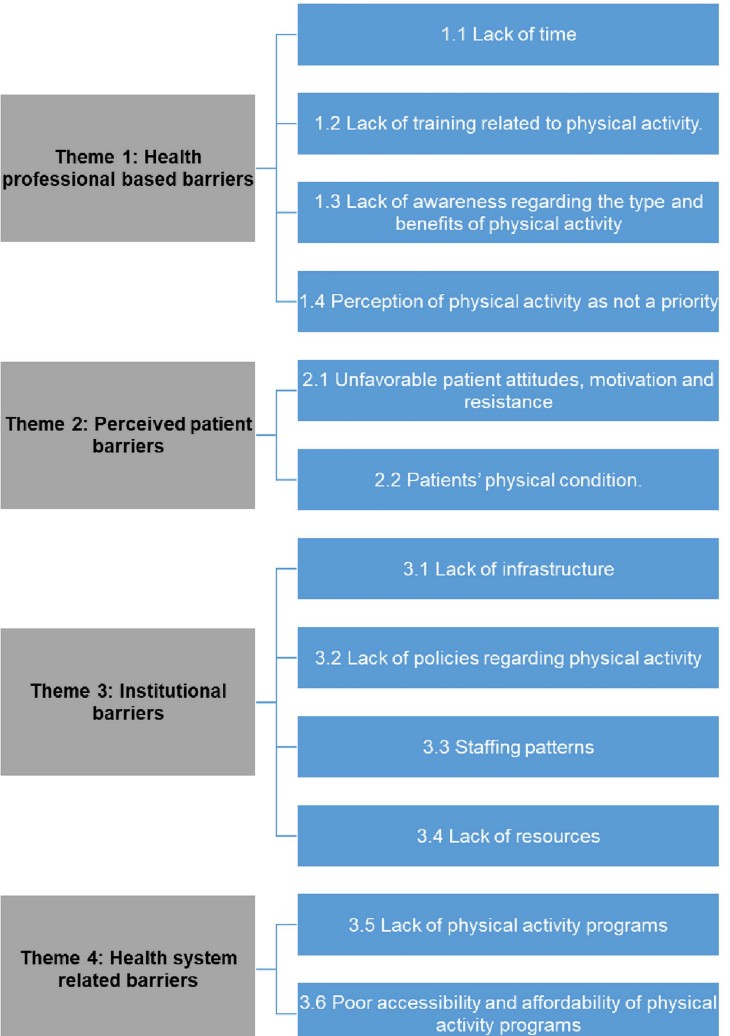

**Lorem Ipsum**

Lorem ipsum dolor sit amet, consectetur adipiscing elit. Mauris maximus fringilla ligula, in malesuada erat tempor ac. Quisque dapibus posuere turpis, vel aliquam massa vehicula non.

**Fig 2. Barriers to exercise promotion.**

It was interesting to note that the promotion of physical activity was not perceived as a priority by some respondents. For example, two studies [32,41] reported negative attitudes of HCPs toward exercise. Similarly, two further studies [20,29] reported that HCPs deemed that promotion of physical activity was not a part of the professional role.

2.  Perceived patient-related barriers

Five studies reported that the patient's unfavourable attitude, including a lack of motivation, lack of interest and resistance, was barrier to actively undertaking physical activity [27,33,40,42,44]. Because the patients' physical condition is associated with the ability to do physical activity, three studies reported that patients' poor health status, being confined to bed rest, or being in pain, all hindered effective physical activity training [31,33,44].

3. Organisational barriers

Several barriers were identified in relation to the organisation itself, where the HCPs worked. Lack of infrastructure, including space, equipment and furniture, was reported as a barrier in 3 studies [9,12,31]. Another barrier was the lack of policies, procedures, or protocols in relation to the promotion of physical activities [19,29,34]. Furthermore, a lack of resources, such as educational pamphlets or materials, was identified in two studies [21,40]. The results also reflect that staffing was a barrier, as an inadequate staff-to-patient ratio increased the workload of staff members and thereby hindered physical activity promotion [9,13,21].

4. Health system-related barriers

Most studies identified some barriers related to the health system. This type of barrier was mainly due to a lack of available exercise programs, including community-based rehabilitation programs for patients [14,27,28]. Other barriers identified were poor access and affordability of exercise programs [27,29]. Indeed, a lack of funding and the non-reimbursement of physical activity counselling hindered the accessibility and affordability of physical activity programs [30].

**Secondary outcomes. Types of physical activity:** The types of physical activities recommended in studies included aerobic exercises such as walking and jogging [18,37]; cardiovascular physical activity and weight training [18,40]; flexibility exercises and lifestyle modification [37]. A further study advised patients to do activities like running, jumping, lifting, cycling, swimming, jogging, and climbing, depending on the physical condition and activity level of the patient [27].

**Data collection techniques:** Various forms of data collection methods were employed to assess HCPs' attitudes, behaviours, and barriers toward exercise promotion among patients. Focus group discussions were used by four studies [11–13,42], whereas eight studies used interviews [9,20,17,22,23,29,32,43]. Twenty-two studies used different types of questionnaires. However, not all of these studies named the questionnaire employed. Of the studies reviewed, four employed questionnaires based on Ajzen's Theory of Planned Behaviour [11,40,50–52]. One study developed the 'Primary Care Staff Views and Experience Survey' [37], which was subsequently used in two additional studies [27,28]. Finally, the Portuguese version of The Exercise in Mental Illness Questionnaire (EMIQ) was used in one study [18].

**Results for the quality appraisal of the included studies.** Tables 2 and 3 present a summary of the CASP assessments conducted for the included studies. Of the 33 included studies, 12 were evaluated as high quality, 18 were rated as moderate quality, and 4 were classified as low quality. The average critical appraisal score for all studies was 80.4 (SD:±0.13). The cohort studies had a range of quality, including low, moderate, and high scores, with an average score of 75.9% (SD:±0.11). Conversely, the qualitative studies demonstrated moderate to high quality, with an average score of 93% (SD:±0.08).

## 5. Discussion

This review provides a synthesis of the literature exploring health care professionals' attitudes, behaviours, and barriers toward exercise promotion among patients. Thirty-four quantitative and qualitative research studies met the inclusion criteria and formed the basis of this review. Overall, HCPs demonstrated positive attitudes toward promoting PA, viewing it as an integral part of their professional responsibilities. Furthermore, they reported confidence in advising and promoting PA to their patients.

Many HCPs in the selected studies believed that physical activity could promote health, improve quality of life, reduce fatigue, manage side effects of treatments, facilitate behavioural changes, lower cardiovascular risk, and improve activity tolerance. They acknowledged both the benefits and potential risks of PA, emphasising the need to tailor recommendations to the individual patient's clinical condition and abilities. While attitudes toward PA were generally positive, some studies highlighted that certain HCPs did not consider PA promotion a priority or an essential part of their professional role [9,11,20,29]. The attitudes of HCPs—whether positive or negative—significantly influence their practices. HCPs with positive attitudes and confidence in recommending PA are more likely to integrate it into patient care and model such behaviors themselves.

**Table 2. Critical appraisal of cohort studies.**

| Author | Criteria | | | | | | | | | | | | | | Score | Total |
|---|---|---|---|---|---|---|---|---|---|---|---|---|---|---|---|---|
| | 1 | 2 | 3 | 4 | 5a | 5b | 6a | 6b | 7 | 8 | 9 | 10 | 11 | 12 | | |
| Al Gamdi 2017 | Y | Y | Y | Y | Y | N | Y | Y | I | I | Y | Y | Y | Y | 11/12; 92% | H |
| Aldosarry 2012 | Y | N/C | Y | Y | N | N | N/C | N/C | I | I | Y | Y | Y | Y | 7/12; 58% | L |
| Cantwell 2017 | Y | Y | Y | Y | Y | N | Y | Y | I | I | Y | Y | Y | Y | 11/12; 92% | H |
| Cunningham; 2021 | Y | Y | Y | Y | Y | N | Y | N | I | I | Y | Y | Y | Y | 10/12; 83% | M |
| Dermody 2017 | Y | N/C | Y | Y | N | N | N/C | N/C | I | I | Y | Y | Y | Y | 7/12; 58% | L |
| Esposito 2011 | Y | Y | Y | Y | Y | N | N/C | N/C | I | I | Y | Y | Y | Y | 9/12; 75% | M |
| Goodman 2011 | Y | N/C | Y | Y | N | N | N/C | N/C | I | I | Y | Y | Y | Y | 7/12; 58% | L |
| Hardcastle 2017 | Y | Y | Y | Y | Y | N | Y | Y | I | I | Y | Y | Y | Y | 11/12; 92% | H |
| Hausmann 2018 | Y | Y | Y | Y | N | N | Y | Y | I | I | Y | Y | Y | Y | 10/12; 83% | M |
| Karvinen 2012 | Y | Y | Y | Y | N | N | N/C | N/C | I | I | Y | Y | Y | Y | 8/12; 67% | M |
| Keogh 2017 | Y | Y | Y | Y | N | N | Y | Y | I | I | Y | Y | Y | Y | 10/12; 83% | M |
| Kleemann 2020 | Y | Y | Y | Y | Y | N | N/C | N/C | I | I | Y | Y | Y | Y | 9/12; 75% | M |
| Leemrijse 2015 | Y | Y | Y | Y | N | N | N/C | N/C | I | I | Y | Y | Y | Y | 8/12; 67% | M |
| McDowell 1997 | Y | Y | Y | Y | Y | N | N/C | N/C | I | I | Y | Y | Y | Y | 9/12; 75% | M |
| Nadler 2017 | Y | Y | Y | Y | Y | N | N/C | N/C | I | I | Y | Y | Y | Y | 9/12; 75% | M |
| Nease 2021 | Y | Y | Y | Y | N | N | N/C | N/C | I | I | Y | Y | Y | Y | 8/12; 67% | M |
| Park 2015 | Y | Y | Y | Y | Y | N | N/C | N/C | I | I | Y | Y | Y | Y | 9/12; 75% | M |
| Pearson 2017 | Y | Y | Y | Y | Y | N | N/C | N/C | I | I | Y | Y | Y | Y | 9/12; 75% | M |
| Shirley, 2001 | Y | Y | Y | Y | Y | N | N/C | N/C | I | I | Y | Y | Y | Y | 9/12; 75% | M |
| Spellman 2014 | Y | Y | N | N | Y | N | N/C | N/C | I | I | Y | Y | Y | Y | 7/12; 58% | L |
| Ungar 2019 | Y | Y | Y | Y | Y | N | Y | Y | I | I | Y | Y | Y | Y | 11/12; 92% | H |

Y – Yes; N – No; N/C – Not clear; I: Indicated in the study, H – High Quality (100%−90%); M – Moderate Quality (89%−61%); L – Low Quality (60%−0)

**Table 3. Critical appraisal of qualitative studies.**

| Author | Criteria | | | | | | | | | | Score | Total |
|---|---|---|---|---|---|---|---|---|---|---|---|---|
| | 1 | 2 | 3 | 4 | 5 | 6 | 7 | 8 | 9 | 10 | | |
| Avancini 2021 | Y | Y | Y | Y | Y | Y | N/C | Y | Y | I | 8/9; 89% | M |
| Boltz 2011 | Y | Y | Y | Y | Y | Y | Y | Y | Y | I | 9/9; 100% | H |
| Chan 2019 | Y | Y | Y | Y | Y | Y | Y | Y | Y | I | 9/9; 100% | H |
| De Klein 2021 | Y | Y | Y | Y | Y | Y | Y | Y | Y | I | 9/9; 100% | H |
| Din 2015 | Y | Y | Y | Y | Y | Y | N | Y | Y | I | 8/9; 89% | M |
| Doherty-King 2011 | Y | Y | Y | Y | Y | N/C | N | Y | Y | I | 7/9; 78% | M |
| Doherty-King 2013 | Y | Y | Y | Y | Y | N/C | N | Y | Y | I | 7/9; 78% | M |
| Douglas 2006 | Y | Y | Y | Y | Y | Y | Y | Y | Y | I | 9/9; 100% | H |
| Jenna 2016 | Y | Y | Y | Y | Y | Y | Y | Y | Y | I | 9/9; 100% | H |
| Kime 2020 | Y | Y | Y | Y | Y | Y | N | Y | Y | I | 8/9; 89% | M |
| Verhaeghe 2013 | Y | Y | Y | Y | Y | Y | Y | Y | Y | I | 9/9; 100% | H |
| Williams 2016 | Y | Y | Y | Y | Y | Y | Y | Y | Y | I | 9/9; 100% | H |
| Wu 2012 | Y | Y | Y | Y | Y | Y | Y | Y | Y | I | 9/9; 100% | H |

Y – Yes; N – No; N/C – Not clear; I: Indicated in the study, H – High Quality (100%−90%); M – Moderate Quality (89%−61%); L – Low Quality (60%−0)

Of the 34 studies included, 16 targeted nurses. This is unsurprising, given nurses' critical role in promoting PA for patients with chronic diseases, such as cardiovascular disease, diabetes, cancer, and chronic obstructive pulmonary disease, which are leading causes of morbidity and mortality worldwide. A systematic review including 18 studies revealed that nurses perceive themselves as having an active role in promoting physical activity and consider it to be an important part of their professional responsibilities [47]. A total of 13 targeted the multidisciplinary team. Multidisciplinary teams are essential for promoting and restoring the health of patients in general as they bring together HCPs from different disciplines to collaborate on patient care, provide comprehensive and coordinated care, and determine patients' educational needs [48].

The studies also reported on the types of physical activities recommended by HCPs. These were mainly aerobic exercises such as walking and jogging, running, jumping, lifting, cycling, swimming, and climbing. Several studies reported positive impacts of aerobic exercises, either alone or when combined with other interventions, on various health outcomes of patients with chronic diseases [50,51]. These outcomes include, but are not limited to, glycaemic control, cognitive function, quality of life, and sleep quality. One study, reported improved pain and functional capacity in patients with Fibromyalgia after swimming three times a week for 12 weeks. A 2021 systematic review that included six RCTs noted the benefits of dance/movement therapy in improving mental health and quality of life for breast cancer patients [59]. However, dancing was not commonly recommended by HCPs in the current review's included studies, potentially due to cultural considerations or the keywords used in the review's search strategy. The lack of recommendation of dance may relate to the keywords used in our search and should not be interpreted as a major finding. It is acknowledged that HCPs may exercise caution when recommending dancing, particularly as it may not be culturally accepted in certain contexts. Indeed, two studies emphasised that PA recommendations for patients should be made cautiously, considering individual patient conditions [27,35].

The presence of standardised protocols and guidelines for recommending exercise for patients with chronic conditions is essential to encourage optimal daily self-care management. This review identified organisational barriers as the most frequently reported impediments to PA promotion. These include the lack of infrastructure, policies, educational resources, and appropriate staff-patient ratios. The argument is that evidence-based guidelines for PA may exist, but hospitals fail to adopt them, which can result in HCPs being unaware of their existence. One study reported that 65% of physicians and 30% of nurses were unaware of physical activity guidelines during and after haematological cancer treatment [59]. In another study, the majority of physiotherapists were unaware of the World Health Organisation (WHO) physical activity guidelines for adults and adolescents [49].

This review identified barriers faced by HCPs in promoting PA, notably limited time and insufficient knowledge about different types and benefits of physical activities. These three barriers have been consistently identified as predominant in several systematic reviews focused on synthesising the barriers to managing and implementing care in healthcare settings. Schofield, Rolfe [52] in their systematic review identified time restraints and a lack of knowledge as barriers to undertaking health promotion interventions in urgent and emergency care. Patient-related barriers included poor health status, lack of motivation or interest, and resistance to PA recommendations. These findings, based on HCP perspectives, warrant further investigation from patients' viewpoints to gain a more comprehensive understanding of these barriers. Furthermore, patients' poor health status, lack of motivation, lack of interest, and resistance from patients were the main perceived patient-related barriers to physical activities. These findings should be interpreted with caution as they represent the perspectives of HCPs regarding patients. To gain a more comprehensive understanding, further research should explore the views of patients regarding the barriers to engaging in PA that are recommended by HCPs.

Overall, the results of this review highlight the importance of a multifaceted approach in addressing barriers to PA recommendations. By understanding the complex interplay between healthcare professionals' attitudes, the existing organisational structures, and patient perceptions, we can better facilitate the promotion of physical activity among patients. Future studies should aim to develop and evaluate interventions that not only educate HCPs but also engage patients in the process of physical activity promotion, considering their individual needs and cultural contexts.

The CASP assessments conducted for the included studies provided valuable insights into the methodological quality of the evidence base. Our findings revealed a considerable variation in the quality of the studies, with 12 out of 34 studies evaluated as high quality, 18 as moderate quality, and 4 classified as low quality. This variation underscores the importance of critically appraising the included studies to assess their reliability and potential impact on the overall findings. The average critical appraisal score of 80.4 (SD: ±0.13) indicates an overall satisfactory level of methodological quality in the included studies. However, it is crucial to acknowledge that certain studies demonstrated limitations and areas for improvement. By identifying these areas, we can better understand the strengths and weaknesses of the evidence.

In terms of study design, the cohort studies exhibited a range of quality, with scores spanning from low to moderate to high, and an average score of 75.9% (SD: ±0.11). The common problems identified in this subset of studies included inadequate consideration of confounding variables and insufficient follow-up of subjects. These issues highlight the need for improved study design and rigorous data collection practices to enhance the validity and reliability of cohort studies in our research area. On the other hand, the qualitative studies demonstrated moderate to high quality, with an average score of 93% (SD: ±0.08). This indicates the robustness of the qualitative evidence base and the credibility of the findings. The comprehensive attention to ethical considerations in these studies enhances the trustworthiness of the qualitative evidence. While our findings shed light on the strengths and limitations of the included studies, it is important to acknowledge the inherent limitations of the research. The evaluation of study quality is subjective to some extent, as it relies on the judgment and expertise of the reviewers. Furthermore, the identified limitations in the included studies may introduce bias and impact the generalisability of the findings. Based on our assessment results, future research should aim to address the identified limitations, particularly in cohort studies, by implementing rigorous study designs, considering confounding factors, and ensuring adequate follow-up of subjects. Further qualitative research is also encouraged to provide in-depth insights into the phenomenon under investigation.

### 5.1. Limitations

This review has several limitations. Firstly, only papers published in the English language were included. Additionally, there was significant heterogeneity across the selected studies, including diverse study methods, assessment tools, and attitudes measured. The attitudes in the quantitative studies included were measured by twenty different instruments, all measuring attitudes in various ways. Additionally, the way questions are phrased or framed can influence how participants respond, leading to response bias and inaccurate measurements of attitudes. Social desirability bias may also be a factor, as HCPs may respond in a socially desirable way, rather than expressing their true attitudes. Attitudes can also vary across different cultures, which can make it challenging to develop instruments that accurately measure attitudes across diverse health professional populations. Lastly, attitudes can change over time, which can make it challenging to accurately measure attitudes using instruments that were developed for different time periods, types of physical activities, and patients.

Most of the quantitative studies used a cross-sectional survey design, which might limit generalisability due to small sample sizes, sampling methods, and limited control over confounding variables [18,33,49]. Finally, the included papers were obtained from a wide range of countries with different healthcare systems and cultures, which may also limit the generalisability of the findings.

### 6. Conclusion

This systematic review underscores the positive attitudes of healthcare professionals toward promoting physical activity, emphasising its role in enhancing physical well-being, preventing complications, and supporting patient progress. However, it also highlights numerous barriers to the promotion of physical activity, including healthcare professional-related, patient-related, organisational, and systemic challenges. Addressing these barriers requires a multifaceted strategy that includes fostering a supportive environment through policy development, adequate resource allocation, and recognition of

healthcare professionals' contributions. Engaging patients is equally critical. Educating them about the benefits of physical activity, collaboratively setting exercise goals, and monitoring their progress can improve adherence and enhance health outcomes. Future research should focus on evaluating the effectiveness of these strategies, exploring innovative approaches to overcoming barriers, and determining the extent to which promotion and adherence to physical activity can be improved across diverse healthcare contexts.

## Supporting information

**S1 Table. PRISMA 2020 checklist.**
(DOCX)

**S2 Table. Characteristics of included studies.**
(DOCX)

## Author contributions

**Conceptualization:** Pinar Avsar, Zena Moore, Husain Nasaif, Barry Moore, Declan Patton, Vishnu Renjith.

**Data curation:** Pinar Avsar, Barry Moore, Tom O'Connor, Vishnu Renjith.

**Methodology:** Pinar Avsar.

**Supervision:** Zena Moore.

**Writing – original draft:** Pinar Avsar, Zena Moore, Husain Nasaif, Barry Moore, Declan Patton, Tom O'Connor, Vishnu Renjith.

**Writing – review & editing:** Pinar Avsar, Zena Moore, Husain Nasaif, Declan Patton, Tom O'Connor, Vishnu Renjith.

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
