## [Decision Letter · Decision Letter 0]

5 Sep 2024

Dear Dr. Avsar,

Thank you for submitting your manuscript to PLOS ONE. After careful consideration, we feel that it has merit but does not fully meet PLOS ONE’s publication criteria as it currently stands. Therefore, we invite you to submit a revised version of the manuscript that addresses the points raised during the review process.

We look forward to receiving your revised manuscript.

Kind regards,

Mr. Nuru Abdu

Academic Editor

PLOS ONE

**Journal Requirements:**

-The economic impact of pressure ulcers among patients in

intensive care units. A systematic review (DOI: 10.1016/j.jtv.2020.12.004)

In your revision ensure you cite all your sources (including your own works), and quote or rephrase any duplicated text outside the methods section. Further consideration is dependent on these concerns being addressed.

5. As required by our policy on Data Availability, please ensure your manuscript or supplementary information includes the following: 

**Additional Editor Comments:**

1. English editing required. Coherence and readability of the manuscript need improvement.

2. Your manuscript does not adhere to PLOS ONE manuscript preparation guidelines. Use the following link to access the manuscript preparation guideline https://learn.inasp.info/pluginfile.php/159252/mod_folder/content/0/PLOS_ONE_manuscript_guidelines_saved.pdf?forcedownload=1

Reviewers' comments:

Reviewer's Responses to Questions

**Comments to the Author**

1. Is the manuscript technically sound, and do the data support the conclusions?

Reviewer #1: Partly

Reviewer #2: Yes

2. Has the statistical analysis been performed appropriately and rigorously?

Reviewer #1: N/A

Reviewer #2: N/A

3. Have the authors made all data underlying the findings in their manuscript fully available?

Reviewer #1: Yes

Reviewer #2: Yes

4. Is the manuscript presented in an intelligible fashion and written in standard English?

Reviewer #1: Yes

Reviewer #2: No

**Reviewer #1:**  Revision of Manuscript Number: PONE-D-24-13967

Abstract:

Clear and nicely written abstract. However, I think the conclusion section sounds like list of recommendations than summary of the striking data. So please revise.

Introduction:

No need to include the PICO and research question. Probably the primary and secondary desired outcomes can be summarised in the aim part.

Methods:

I have concerns about the keywords used, for example “behavior” is a key word in the title but not found in the search strategy. Also, you should either use HCPs in general or list all related specialties involved not only physicians and nurses. Someone can argue physiotherapists are more attached to this than nurses, pharmacists ….etc.

It would be useful if you provide the results of the search process on one of the databases as a supplementary (PubMed or Scopus).

The manuscript needs language revision for better presentation.

Data analysis section needs extensive revision as I was confused with why meta analysis is essentially mentioned in such type of review with mostly qualitative data (heterogeneity of what? Normally we use this term in studies with quantitative data in nature). How did you do the coding? Is it content analysis or thematic or sentimental ?..... etc

Results:

Where is figure 1? Why did you provide Table 1 with only 3 studies?

In Table 2 you need to add the publication year next to authors and a column for quality assessment.

It was very abrupt to see “Data collection techniques” in the results section page 22! Why here ?

How did you dealt with data from weaker or incomplete studies?

Discussion:

There are parts where new results were presented for the first time this shouldn't be the case, please revise. The overall discussion needs more depth and critical evaluation. Readers want to see your input for why do you think these were the outcomes of the review.

**Reviewer #2:**  The study topic is of interest and highlights crucial points in the health promotion program. However, there certain limitations that should be addressed:

1. Within the electronic search section specify if there were no filters or limits used,

2. The keyword used in the search process are limited, justify the limited used of keywords for example, physiotherapist or clinician is not included. The keywords exercise or physical activity are also limited as synonyms such as aerobics or movement therapy are not included.

3. Provide operational definitions for HCPs and exercises, as it will simplify the issues concerning selection of keywords.

4. In the study selection section specify that studies in English language were used

5. The initials of the reviewers who did the first screening of the studies are not mentioned in the study selection section

6. On page 8, amend the sentence “… the themes and categories presented under the sections of attitudes, practices, and behaviors.” As “…. attitudes, behaviors and barriers.” As practice was not assessed.

7. Amend the table referring to the CASP assessments to table 3, on page 9.

8. The criteria used in the CASP assessments should be described in the methods section or provided as footnotes in the tables.

9. On page 10, use consistent citation style as per the PLOS One referencing guidelines.

10. I recommend table 2, the characteristics of the studies used to be handled as supplementary file, as it increases the bulkiness of the manuscript without added benefit.

11. How did you manage to avoid bias from the themes already presented in the quantitative questionnaires? As discussed in the limitations, the way questions are phrased or framed can influence how participants respond, leading to response bias and inaccurate measurements of attitudes and themes generated. The same applies to behaviors and barriers, justify.

12. The abbreviations SD and the presented figures are not consistent, if you are referring to SD should be presented in integers not percentages as percentage refers to relative standard deviation (RSD).

13. In the discussion section, page 27, the issue that dancing was not recommended to patients by HCP may be due to the keywords and should not be discussed as a major result.

14. The manuscript needs English language editing.

**Do you want your identity to be public for this peer review?** For information about this choice, including consent withdrawal, please see our Privacy Policy

Reviewer #1: No

Reviewer #2: No

---

## [Author Response · Author response to Decision Letter 1]

22 Oct 2024

Dear Editor,

We would like to thank the reviewer for careful and thorough reading of this manuscript and for the thoughtful comments and constructive suggestions, which help to improve the quality of this manuscript for publication.

We believe that we have carefully addressed all the comments. Should you have any additional requests or questions, please do not hesitate to contact me.

Yours sincerely,

Pinar Avsar

Reviewer Comment

Answered Page

Review report Within the electronic search section specify if there were no filters or limits used,

We have clarified the details regarding the use of filters and limits in the electronic search section of our manuscript. 7

The keyword used in the search process are limited, justify the limited used of keywords for example, physiotherapist or clinician is not included. The keywords exercise or physical activity are also limited as synonyms such as aerobics or movement therapy are not included.

While appreciating the comment of the respected reviewer, the keywords were selected based on the most commonly used terms in the literature to ensure that the search captures a broad range of studies involving healthcare professionals' perspectives on exercise. We intentionally focused on general terms like "healthcare professionals" rather than more specific terms such as "physiotherapist" to avoid narrowing the scope of the search too much. Additionally, while "exercise" and "physical activity" were chosen as umbrella terms to capture a variety of interventions, specific terms like "aerobics" or "movement therapy" were excluded to maintain a manageable scope and to avoid overly fragmenting the search results. However, these terms were considered during the title and abstract screening phase to ensure relevant studies were not missed.

Provide operational definitions for HCPs and exercises, as it will simplify the issues concerning selection of keywords.

To address your concerns, we have added operational definitions for "Healthcare Professionals (HCPs)" and "exercise" to the manuscript. 8

In the study selection section specify that studies in English language were used

Thanks for the comment. It was added. 7

The initials of the reviewers who did the first screening of the studies are not mentioned in the study selection section

Thank you for your feedback. We have added the initials of the reviewers involved in the study selection process to the manuscript for clarity. 8

On page 8, amend the sentence “… the themes and categories presented under the sections of attitudes, practices, and behaviors.” As “…. attitudes, behaviors and barriers.” As practice was not assessed.

Thank you for your suggestion. We have amended the sentence and practice is removed.

Amend the table referring to the CASP assessments to table 3, on page 9.

Thanks for the comment. We amended it. 9

The criteria used in the CASP assessments should be described in the methods section or provided as footnotes in the tables.

Thank you for your suggestion. We have added a "Quality Appraisal" subheading to the methods section, where we describe the criteria used in the CASP assessments, as recommended. 9

On page 10, use consistent citation style as per the PLOS One referencing guidelines.

Thank you for your suggestion. We have revised the citations on page 10 to ensure consistency with the PLOS ONE referencing guidelines. 10

I recommend table 2, the characteristics of the studies used to be handled as supplementary file, as it increases the bulkiness of the manuscript without added benefit.

Thank you for your recommendation. We have moved Table 2, which outlines the characteristics of the studies. This is added as a supplementary file.

How did you manage to avoid bias from the themes already presented in the quantitative questionnaires? As discussed in the limitations, the way questions are phrased or framed can influence how participants respond, leading to response bias and inaccurate measurements of attitudes and themes generated. The same applies to behaviors and barriers, justify.

Thank you for your insightful comment regarding the potential for bias stemming from the themes presented in the quantitative questionnaires. To mitigate this risk, we employed a multi-faceted approach:

Diverse Sources: We ensured that the findings are derived from a variety of sources, including interviews and focus groups, in addition to the quantitative questionnaires. This allowed us to capture a broader range of perspectives and minimize reliance on any single data source.

Thematic Analysis: We conducted a rigorous thematic analysis, where the themes were developed. The emerged themes were reviewed by the research team to ensure the internal validity. This process strived to minimize individual biases and led to a more balanced representation of the data.

Reflexivity: To ensure rigor we acknowledged our own biases and preconceptions during the data analysis process, actively reflecting on how these might influence the interpretation of the findings.

We recognize that response bias can affect the accuracy of measurements related to attitudes, behaviors, and barriers, and we have highlighted these limitations in our discussion. However, we believe that the measures taken have strengthened the validity of our findings.

The abbreviations SD and the presented figures are not consistent, if you are referring to SD should be presented in integers not percentages as percentage refers to relative standard deviation (RSD).

Thank you for your feedback. We have revised the manuscript to ensure consistency in the presentation of the standard deviation (SD), now presenting it as integers rather than percentages. Whole document

In the discussion section, page 27, the issue that dancing was not recommended to patients by HCP may be due to the keywords and should not be discussed as a major result.

Thank you for your feedback. We have revised the discussion section. 28

The manuscript needs English language editing.

The manuscript has been edited for English language. Whole document.

Reviewer #1 Abstract:

Clear and nicely written abstract. However, I think the conclusion section sounds like list of recommendations than summary of the striking data. So please revise.

Thank you for your comment. The conclusion section was revised. 1

Introduction:

No need to include the PICO and research question. Probably the primary and secondary desired outcomes can be summarised in the aim part. Thank you for your suggestion. We have integrated the primary and secondary desired outcomes into the 'Aim' section to provide a more concise overview of the study's focus, as recommended. 8

Methods:

I have concerns about the keywords used, for example “behavior” is a key word in the title but not found in the search strategy. Also, you should either use HCPs in general or list all related specialties involved not only physicians and nurses. Someone can argue physiotherapists are more attached to this than nurses, pharmacists ….etc.

It would be useful if you provide the results of the search process on one of the databases as a supplementary (PubMed or Scopus). Thank you for your valuable feedback. We have updated our search strategy to include the keyword 'behavior' to ensure consistency with the title and the scope of the study.

While we appreciate the insightful comment from the respected reviewer, the decision to use the general term 'healthcare professionals' rather than listing specific specialties such as 'pharmacists' was intentional. This approach aimed to capture a broad range of studies and perspectives without unduly restricting the scope of the search. Using a more general term allowed us to include various healthcare professionals involved in exercise promotion, such as physicians and nurses, ensuring that relevant studies from different disciplines were considered. During the title and abstract screening phase, we carefully evaluated the relevance of each study, including those focusing on specific groups like physiotherapists, pharmacist to ensure that significant contributions were not overlooked.

The manuscript needs language revision for better presentation.

It was added. Whole document.

Data analysis section needs extensive revision as I was confused with why meta analysis is essentially mentioned in such type of review with mostly qualitative data (heterogeneity of what? Normally we use this term in studies with quantitative data in nature). How did you do the coding? Is it content analysis or thematic or sentimental ?..... etc Thank you for your valuable comments. We agree that the term 'meta-analysis' was not appropriate given the qualitative nature of much of the data. As such, we have removed the reference to meta-analysis and clarified that a narrative analysis was used. This narrative approach allowed us to accommodate the diverse methodologies and outcomes of the included studies. We employed a qualitative coding process, which involved creating codes, grouping them into categories, and clustering them into themes. We have updated the manuscript to better explain our analysis process. 10

Results:

Where is figure 1? Why did you provide Table 1 with only 3 studies? Thank you for your comments and for bringing this to our attention. Figure 1 has been provided as a separate file in accordance with the journal's submission requirements. The figure outlines the PRISMA flow diagram, detailing the study selection process, including the number of studies screened, excluded, and included in the review.

Regarding Table 1, we included the details of only three studies to provide a concise illustration of the type of data extracted and analyzed in this review. We believed it was a systematic review requirement to present a summary of excluded studies to enhance transparency and clarify the reasons for exclusion.

In Table 2 you need to add the publication year next to authors and a column for quality assessment.

Thank you for your valuable suggestion regarding the addition of the publication year next to the authors' names and including a column for quality assessment in Table 2. We used EndNote for reference management as required by the journal, and the system's formatting limitations did not allow us to automatically display the publication year directly alongside the authors in the table.

It was very abrupt to see “Data collection techniques” in the results section page 22! Why here ? Thank you for your observation. The section on 'Data collection techniques' in the results section refers to the methods and approaches used in the included studies, rather than the data collection process for our systematic review. As such, we felt it was most appropriate to include this information in the results section to provide a comprehensive summary of the methodologies used across the studies we reviewed. This placement aims to give readers a better understanding of how data were gathered in the studies included in our analysis.

How did you dealt with data from weaker or incomplete studies? Thank you for your insightful question. In our systematic review, we adopted a rigorous approach to assess the quality of the included studies using the Critical Appraisal Skills Programme (CASP) tool. For studies identified as weaker or incomplete, we carefully considered their findings in the context of the overall evidence base.

We aimed to extract relevant data while noting any limitations or biases present in these studies. Where applicable, we highlighted the potential impact of these limitations on our findings and discussions.

Discussion:

There are parts where new results were presented for the first time this shouldn't be the case, please revise. The overall discussion needs more depth and critical evaluation. Readers want to see your input for why do you think these were the outcomes of the review. We have revised the discussion to enhance its depth and critical evaluation. We have also ensured that all presented results align with those previously reported in the literature, avoiding any new findings being introduced. Discussion section

Reviewer #2 Reviewer 2's comments overlapped with the review report. Therefore, we have addressed this in our response above.

---

## [Decision Letter · Decision Letter 1]

8 Dec 2024

Dear Dr. Avsar,

Thank you for submitting your manuscript to PLOS ONE. After careful consideration, we feel that it has merit but does not fully meet PLOS ONE’s publication criteria as it currently stands. Therefore, we invite you to submit a revised version of the manuscript that addresses the points raised during the review process.

We look forward to receiving your revised manuscript.

Kind regards,

Nuru Abdu, BPharm

Academic Editor

PLOS ONE

Journal Requirements:

Additional Editor Comments:

1. Massive English Editing required. Coherence and readability of the manuscript need an improvement.

2. The style of the reference citations should be consistent throughout the manuscript and avoid over-citations where applicable.

Reviewers' comments:

Reviewer's Responses to Questions

**Comments to the Author**

Reviewer #1: (No Response)

Reviewer #2: All comments have been addressed

2. Is the manuscript technically sound, and do the data support the conclusions?

Reviewer #1: Yes

Reviewer #2: Yes

3. Has the statistical analysis been performed appropriately and rigorously?

Reviewer #1: N/A

Reviewer #2: N/A

4. Have the authors made all data underlying the findings in their manuscript fully available?

Reviewer #1: Yes

Reviewer #2: Yes

5. Is the manuscript presented in an intelligible fashion and written in standard English?

Reviewer #1: No

Reviewer #2: Yes

Reviewer #1: Please revise the manuscript again . I will put here one paragraph only and comment on the issues in it:

“Geographical location A wide spread of countries are represented in the studies, with the USA having 8 studies(27, 30, 32, 33, 35, 39, 44, 46), the UK having 5 studies (Douglas et al., 2006; Goodman et al., 2011; Kime et al., 2020; Mc Dowell et al., 1997; Williams et al., 2016). Australia, Canada, Germany, The Netherlands, and Saudi Arabia having 2 studies each (17, 24, 25, 29, 38, 41, 43, 47-49). The remaining studies were from Australia and New Zealand (16), Belgium (50), Brazil (14), Ireland (20), Italy (26), Korea (45), Singapore (28), Taiwan (51), The UK and Ireland (15) and Wales (31). One study included people from seven international oncology societies (37).”

• Different referencing style.

• “Australia, Canada, Germany, The Netherlands, and Saudi Arabia having 2 studies each (17, 24, 25, 29, 38, 41, 43, 47-49)” … one missing reference if each country has 2 papers.

• “The remaining studies were from Australia and New Zealand (16)”.. did you count this as Australian or New Zelandian paper?

Reviewer #2: I have no further comments. However, if the recommendation suggested provided in the first review on the methods used to mitigate the risk of bias stemming from the themes presented in the quantitative questionnaires to be included in the methods section.

**Do you want your identity to be public for this peer review?** For information about this choice, including consent withdrawal, please see our Privacy Policy

Reviewer #1: **Yes: ** Dr Abdullah Al Hamid

Reviewer #2: No

---

## [Author Response · Author response to Decision Letter 2]

31 Dec 2024

Dear Editor,

We would like to thank the reviewer for careful and thorough reading of this manuscript and for the thoughtful comments and constructive suggestions, which help to improve the quality of this manuscript for publication.

We believe that we have carefully addressed all the comments. Should you have any additional requests or questions, please do not hesitate to contact me.

Yours sincerely,

Pinar Avsar

Reviewer Comment

Answered Page

Additional Editor Comments

Massive English Editing required. Coherence and readability of the manuscript need an improvement. We have thoroughly revised the manuscript to enhance its language, clarity, and overall structure. The introduction, discussion, and limitations sections were particularly refined to ensure they are more coherent and easier to follow. Whole document

The style of the reference citations should be consistent throughout the manuscript and avoid over-citations where applicable. Thank you for your feedback. We have revised the manuscript to ensure that the style of reference citations is consistent throughout and have minimized over-citations where applicable. Whole document

Reviewers Please revise the manuscript again . I will put here one paragraph only and comment on the issues in it:

“Geographical location A wide spread of countries are represented in the studies, with the USA having 8 studies(27, 30, 32, 33, 35, 39, 44, 46), the UK having 5 studies (Douglas et al., 2006; Goodman et al., 2011; Kime et al., 2020; Mc Dowell et al., 1997; Williams et al., 2016). Australia, Canada, Germany, The Netherlands, and Saudi Arabia having 2 studies each (17, 24, 25, 29, 38, 41, 43, 47-49). The remaining studies were from Australia and New Zealand (16), Belgium (50), Brazil (14), Ireland (20), Italy (26), Korea (45), Singapore (28), Taiwan (51), The UK and Ireland (15) and Wales (31). One study included people from seven international oncology societies (37).”

• Different referencing style. We have carefully reviewed and edited all references in the manuscript to ensure they adhere to a consistent referencing style format. Page 11

• “Australia, Canada, Germany, The Netherlands, and Saudi Arabia having 2 studies each (17, 24, 25, 29, 38, 41, 43, 47-49)” … one missing reference if each country has 2 papers. Thank you for identifying the discrepancy in the references. The section has been revised to accurately reflect that Australia, Canada, Germany, The Netherlands, and Saudi Arabia each have two studies, with the correct references now cited as (17, 27, 28, 32, 41, 44, 46, 50-52). Page 11

“The remaining studies were from Australia and New Zealand (16)”.. did you count this as Australian or New Zelandian paper? Thank you for highlighting this point. The study referenced as (16) was conducted jointly in both Australia and New Zealand, so it was not attributed exclusively to either country. The revised text now clarifies this to avoid any ambiguity. Page 11

Reviewer #2: I have no further comments. However, if the recommendation suggested provided in the first review on the methods used to mitigate the risk of bias stemming from the themes presented in the quantitative questionnaires to be included in the methods section.

Thank you for your valuable suggestion. We have revised the manuscript accordingly and included a detailed subsection under the "Data Analysis" subheading in the Methods section to address this. 9

---

## [Editor Report · Decision Letter 2]

5 Feb 2025

Dear Dr. Avsar,

Thank you for submitting your manuscript to PLOS ONE. After careful consideration, we feel that it has merit but does not fully meet PLOS ONE’s publication criteria as it currently stands. Therefore, we invite you to submit a revised version of the manuscript that addresses the points raised during the review process.

We look forward to receiving your revised manuscript.

Kind regards,

Nuru Abdu, BPharm

Academic Editor

PLOS ONE

Journal Requirements:

Additional Editor Comments:

1. Massive English editing required. Coherence and readability of the manuscript need improvement.

2. Your manuscript does not follow PLOS ONE manuscript preparation guidelines.

---

## [Author Response · Author response to Decision Letter 3]

17 Feb 2025

Dear Editor,

We would like to thank the reviewer for careful and thorough reading of this manuscript and for the thoughtful comments and constructive suggestions, which help to improve the quality of this manuscript for publication.

We believe that we have carefully addressed all the comments. Should you have any additional requests or questions, please do not hesitate to contact me.

Yours sincerely,

Pinar Avsar

Reviewer Comment

Answered Page

Additional Editor Comments Massive English editing required. Coherence and readability of the manuscript need improvement..

Thank you for your feedback. We have revised the manuscript for improved clarity, coherence, and readability to ensure the content is more accessible and aligns with the expectations of the review process. All document

Your manuscript does not follow PLOS ONE manuscript preparation guidelines Thank you for your feedback. We have carefully edited the manuscript to align with the PLOS ONE manuscript preparation guidelines and ensure compliance with all required formatting and structural specifications. 8

Journal Requirements Please review your reference list to ensure that it is complete and correct. If you have cited papers that have been retracted, please include the rationale for doing so in the manuscript text, or remove these references and replace them with relevant current references. Any changes to the reference list should be mentioned in the rebuttal letter that accompanies your revised manuscript. If you need to cite a retracted article, indicate the article’s retracted status in the References list and also include a citation and full reference for the retraction notice. We conducted a thorough search using the Retraction Watch Database, PubMed, and the publishers' official websites. None of the cited references in our manuscript have been retracted.

---

## [Editor Report · Decision Letter 3]

14 Mar 2025

Dear Dr. Avsar,

Thank you for submitting your manuscript to PLOS ONE. After careful consideration, we feel that it has merit but does not fully meet PLOS ONE’s publication criteria as it currently stands. Therefore, we invite you to submit a revised version of the manuscript that addresses the points raised during the review process.

We look forward to receiving your revised manuscript.

Kind regards,

Nuru Abdu, BPharm

Academic Editor

PLOS ONE

Journal Requirements:

Additional Editor Comments:

1. Your manuscript does not follow PLOS ONE manuscript preparation guidelines

2. Your citation does not follow PLOS ONE manuscript

3. Further English editing required to improve the coherence and readability of the manuscript

---

## [Author Response · Author response to Decision Letter 4]

27 Apr 2025

Dear Editor,

We would like to thank the reviewer for careful and thorough reading of this manuscript and for the thoughtful comments and constructive suggestions, which help to improve the quality of this manuscript for publication.

We believe that we have carefully addressed all the comments. Should you have any additional requests or questions, please do not hesitate to contact me.

Yours sincerely,

Pinar Avsar

Reviewer Comment

Answered Page

Editor Comments • The manuscript does not follow the PLOS ONE manuscript preparation guidelines.

• The citation style does not comply with PLOS ONE requirements.

• Further English editing is required to improve the coherence and readability of the manuscript.

Thank you for your constructive feedback. In response to your comments, we have thoroughly revised the manuscript to ensure full compliance with the PLOS ONE manuscript preparation guidelines. The citation style has been corrected throughout the text to align with PLOS ONE referencing requirements. Additionally, the manuscript has undergone comprehensive English editing to improve coherence, grammar, and overall readability. We hope that these revisions address your concerns and enhance the clarity and quality of the submission. All document

---

## [Editor Report · Decision Letter 4]

14 May 2025

Dear Dr. Avsar,

Thank you for submitting your manuscript to PLOS ONE. After careful consideration, we feel that it has merit but does not fully meet PLOS ONE’s publication criteria as it currently stands. Therefore, we invite you to submit a revised version of the manuscript that addresses the points raised during the review process.

We look forward to receiving your revised manuscript.

Kind regards,

Nuru Abdu, BPharm

Academic Editor

PLOS ONE

Journal Requirements:

Additional Editor Comments:

1. Your manuscript does not follow PLOS ONE manuscript preparation guidelines

2. Your citation does not follow PLOS ONE manuscript

3. Further English editing required to improve the coherence and readability of the manuscript

---

## [Author Response · Author response to Decision Letter 5]

16 Jun 2025

Dear Editor,

We would like to thank you for providing the opportunity to revise the manuscript. There were three comments. We believe that we have carefully addressed all the comments. Your suggestions have helped us to improve the quality of this manuscript. Should you have any additional requests or questions, please do not hesitate to contact me.

No Editorial comments Response from the Author

1 The manuscript does not follow the PLOS ONE manuscript preparation guidelines Thanks for the comment. We have reviewed the manuscript preparation guidelines as provided by PLOS ONE. Thorough modifications were made to fit the manuscript into the journal guidelines.

File format: The manuscript has been prepared in docx format

Font: Standard Arial font has been used throughout the manuscript, font size 12.

Layout and spacing: The manuscript is now double-spaced and formatted in a single-column layout.

Page and line numbers: Page numbers and continuous line numbers have been included as required.

Footnotes: All footnotes have been removed.

Abbreviations: Abbreviations are defined upon first use, used consistently. Abbreviations are given at the end of the manuscript.

2 The citation style does not comply with PLOS ONE requirements. All references have been formatted according to the Vancouver style, in line with PLOS ONE guidelines.

EndNote was used to manage the references.

3 Further English editing is required to improve the coherence and readability of the manuscript. A native language speaker edited the manuscript. We then had the manuscript reviewed by a professional copyeditor. Redundancies were eliminated, sentence restructuring was done as needed, and citations were corrected. This has significantly improved the manuscript.

---

## [Editor Report · Decision Letter 5]

7 Aug 2025

Health care professionals' attitudes, behaviours and barriers toward exercise promotion among patients: A systematic review

PONE-D-24-13967R5

Dear Dr. Avsar,

We’re pleased to inform you that your manuscript has been judged scientifically suitable for publication and will be formally accepted for publication once it meets all outstanding technical requirements.

Kind regards,

Nuru Abdu, BPharm

Academic Editor

PLOS ONE
---

## [Editor Report · Acceptance letter]

PONE-D-24-13967R5

PLOS ONE

Dear Dr. Avsar,

I'm pleased to inform you that your manuscript has been deemed suitable for publication in PLOS ONE. Congratulations! Your manuscript is now being handed over to our production team.

Kind regards,

on behalf of

Mr. Nuru Abdu

Academic Editor

PLOS ONE